# Bioinformatics approach for developing a minimum set of SNP markers for identification of temperate *japonica* rice varieties cultivated in Spain

Ester Sales[1], Julia García-Romeral[2], Concha Domingo[2]*

1 Departamento de Ciencias Agrarias y del Medio Natural, Universidad de Zaragoza, Escuela Politécnica Superior, Huesca, Spain, 2 Departamento del Arroz, Instituto Valenciano de Investigaciones Agrarias, Moncada, Spain

* domingo_concar@gva.es

## Abstract

The use of molecular markers for plant variety identification and protection is increasing. For this purpose, SNP markers have provided a reliable and stable tool for plant genotyping. The availability of small and low-cost SNP panels to accelerate the identification of the cultivated rice varieties should be beneficial for breeders, seed certification entities and rice industry. With the intention of providing of such a facility, we first developed a simple and easy-handle bioinformatics tool based on the widely used and freely available software R to generate small sets of SNPs that can discriminate varieties, by selecting markers from a larger genotyping dataset. By applying this algorithm to data from a previously genotyped collection of temperate *japonica* varieties from different countries, we identified a minimal set of 31 SNPs markers to distinguish 210 varieties. In addition, we used this algorithm to discriminate the 43 most cultivated in Spain rice varieties with minimal sets of 8 SNPs. We then developed and tested 22 Kompetitive Allele-Specific PCR (KASP) assays for the markers included in these panels, and obtained reliable genotype patterns for rice varieties identification. The complete 22 markers panel and the rice genotypes data could offer a useful and low-cost tool for rice breeders and industry to identify varieties and therefore to guarantee the quality of rice. The provided R-based algorithm can be applied to other genomic resources to develop core sets of discriminating markers.

## Introduction

Rice is one of the most consumed crops in the world. Yield and quality, as most morphological and physiological traits, including grain attributes and cooking aptitude, are intrinsic to each variety. In this sense, the use of a particular variety for a specific purpose, as well as the purity of grain batches, are important for farmers and food industry. Identification of varieties is important to avoid fraud, which is becoming widespread, particularly in varieties of high commercial value that can be easily adulterated by mixing or substituting with cheaper lower quality varieties. To protect breeders' rights, as well as those from the food industry and

**Funding:** JGR was funded by the Spanish Ministerio de Ciencia, Innovación y Universidades (https://www.ciencia.gob.es/) (fellowship number PRE2019-089034) Funding information and statemnet was missing: This work was supported by the project IVIA-GVA 52201 from Instituto Valenciano de Investigaciones Agrarias (co-financed by the European Union through the ERDF Program 2021-2027 Comunitat Valenciana). The funders had no role in study design, data collection and analysis, decision to publish, or preparation of the manuscript.

**Competing interests:** The authors have declared that no competing interests exist.

consumers, seed certification offers a guaranty of the quality of seed that includes both identity and purity of the variety (Council Directive 66/402/EEC of 14 June 1966 on the marketing of cereal seed). For the 2021 season, a total of 14803 t of certified seeds from 50 rice varieties were commercialized in Spain (www.mapa.gob.es) while in Italy, rice certified seeds reached 42833 t (https://www.crea.gov.it/home). In addition, fraud is also a problem for brands, as they need to ensure the quality of the rice sacks which depends on the variety. Thus, variety identity and purity are important factors to determine the price of the product, and it is important to ensure the identification of the rice variety and its purity by the detection of possible adulteration by mixtures with lower quality varieties.

Plant varieties have historically been identified according to their appearance or morphological traits, named descriptors, by visual inspection. However, this approach is not suitable when rapid results are required or the whole plant is not available [1]. In the European Union, the Community Plant Variety Office (CPVO), following the International Union for the Protection of New Varieties of Plants (UPOV) guidelines, disposes of a comprehensive technical protocol to evaluate the Distinctness, Uniformity, and Stability of a rice variety, based on visually observable traits, such as plant height, seed colour, flag length and width, ligule shape apex or grain length and width [2]. However, this phenotype-based identification of rice varieties becomes progressively inefficient because morphological traits of commercial plant material become similar, which is derived from narrowed genetic pools as a consequence of adaptation to local areas where the breeding programs are developed. Rice cultivation in the European Union is mainly distributed in the Mediterranean region, and the commercialized varieties belong mostly to the temperate *japonica* genetic group, showing both temperature and photo-period conditions adaptation [3]. Besides plant growth adaptation, European rice varieties need to accomplish unique organoleptic characteristics expected by consumers, according to specific cultural uses, such as those of the traditional Italian and Spanish gastronomies. Distinction of crop varieties is, as mentioned, even more difficult when the plant in the field is not available. This is the case of packed rice as, after classification according to their size, grains from different varieties are almost undistinguishable.

Molecular markers are a useful tool for variety identification that can complement phenotype identification. The use of molecular techniques enables certification authorities to identify the plant variety based on laboratory analysis instead of visual phenotypic examination of the plants in the field [4]. Recently, the European Union has recognized the use of molecular techniques as a supplementary tool for field inspections and control plot testing, as well as post-control, when there are doubts as regards the seed varietal identity, for the purposes of phenotypic examination, regarding the species marketed in its countries [4]. DNA markers are frequently used to analyse crop varieties. Simple Sequence Repeat (SSR) have been used for the identification and certification of Chilean rice [5]. More recently, the International Association for Cereal Science and Technology (ICC) has approved the standard method number 187 for the identification of rice varieties using 12 SSR markers [6]. Among the molecular markers, Single Nucleotide Polymorphisms (SNPs) are becoming widely employed [7, 8]. SNP panels were employed to distinguish 37 varieties in the specific subpopulation of Basmati rice [9], to discriminate 368 Indian wheat varieties [10] or to differentiate 116 accessions of *Humulus lulupus* [11]. SNPs are genomic variations at a single base position, and they are inherited and stable. As a result of resequencing efforts performed during the last years and the development of high density SNP microarrays, there is abundance of crop genomes sequences and variation information in databases [12, 13]. Particularly, rice SNPs are compiled in databases as the Rice SNP-Seek Database (http://snp-seek.irri.org), the Rice Diversity Project database (https://ricediversity.org/), the Rice Genome Annotation Project database (http://rice.plantbiology.msu.edu/), or Gramene database (https://www.gramene.org/). SNPs present advantages

respect to other markers commonly used, as microsatellites, as they are easily detectable trough different technologies as resequencing genomes or Genotyping by Sequencing (GBS). Furthermore, SNPs are easily detected using Kompetitive Allele-Specific PCR (KASP) assays [14]. KASP genotyping assays are based on competitive allele specific PCR. Allele specific primers each harbouring a unique tail sequence enables allele discrimination. The advantages of SNP analysis using KASP assays are that they are cost effective to run, and the assays are simple to set up [7].

This study attends the need of developing an efficient molecular tool for the identification and discrimination of rice varieties cultivated in Southern Europe, to provide assurance of the correct variety, and therefore maintaining the quality standards. In a previous work, we characterized a collection of 217 *japonica* rice varieties cultivated in temperate regions using SNP markers [15]. This collection represents the genetic diversity of cultivated rice in Northern latitudes and can be used to find genetic markers that discriminate among varieties grown in the Mediterranean basin. The collection was fingerprinted using 2094 SNPs distributed along the genome, and the genetic profiles enable the design of sets of markers to identify the varieties individually. To simplify assays and to facilitate a low cost and automatable method, a minimal core PCR-derived marker set that can discriminate between any pair of varieties should be established. Development of minimal KASP marker panels for distinguishing genotypes has been addressed in fruit trees, as apple, and crops including barley and rice [7, 16, 17]. Some algorithms for this purpose have been developed previously [18], but they require bioinformatics skills or paid software. We have developed a bioinformatics algorithm able to identify a minimal set of SNP markers which discriminate most rice varieties cultivated in Spain using the cost-free statistic package R. The simplicity of this algorithm makes it easy to run on standard laboratory-type computer systems with no need of high-capacity servers. Selected markers were tested in a set of 44 rice varieties that are cultivated in Spain. We used rice kernels instead of leaves for preparing DNA samples to validate the methodology for both packed rice and for seed certification.

## Materials and methods

### Plant material

Seeds from 44 rice varieties were used in this study, which are current Spanish or Italian commercial varieties (S1 Table). Seeds were provided by IVIA (Instituto Valenciano de Investigaciones Agrarias) and three seed certification companies, Cooperativa de Productores de Semillas de Arroz (Copsemar), Semillas Certificadas Castells, and Arrocerías Herba.

### Selection of a set of discriminating SNP markers

A previously generated genotype dataset derived from a custom Infinium SNP genotyping array compiling 2094 biallelic SNPs and 217 accessions representative of *japonica* rice cultivated in temperate regions [15] was used in this study to identify SNPs that discriminate the selected plant material.

We developed an algorithm using the R software (http://www.R-project.org) (S2 Table) that was initially applied to this genotypic dataset to discriminate the varieties included in the collection. First, we removed SNPs that showed deletions in some accessions and then, a matrix with 1317 SNPs remained. Second, it was necessary to identify, if any, pairs or groups of varieties that cannot be discriminated by these markers. If such pair of varieties was found, they were treated as a single accession. At this step, the algorithm generated a list of these pairs of non-differentiable accessions (S3 Table). The third step was to perform pairwise comparisons among the genotypes to find the group of markers that discriminate each pair. For these

markers, the algorithm calculates those that differentiate the same pairs of varieties, that is, the redundant markers that are removed (S4 Table). One marker, the first in the list, is selected to proceed to the following step and the remaining are removed for calculations. Finally, pairwise comparisons were performed by iteration, which produced 5 different combinations of 31 SNP markers that distinguished the varieties in the collection (S5 Table).

To find a set of minimal markers to distinguish Spanish rice varieties, we used genotypic data from 27 varieties included in this collection (S1 Table) that are currently cultivated in Spain. First, we removed 186 SNPs that showed deletions in some accessions, and also those that were monomorphic; then a matrix with 1665 SNPs remained. Second, pairs or groups of varieties that cannot be discriminated by these markers were identified. and a list of these pairs of non-differentiable accessions was generated.

In the third step pairwise comparisons among the 27 genotypes were performed to find the group of markers that discriminate each pair and then, the algorithm calculated the redundant markers. Once redundant markers have been discarded, we obtained a set of 780 markers. Finally, pairwise comparisons were performed by iteration, which produced different combinations of markers that distinguished the 27 varieties. The algorithm generated 94 sets of 6 SNPs markers. Since some markers were present in more than one combination, we selected 22 SNPs markers (S6 Table) potentially useful for the identification of rice varieties cultivated in Spain. After validation of the primers, we added 17 varieties to the list that were not previously genotyped, and generated a matrix for the unequivocally identification of the 44 rice varieties which was used as input in our previously designed script pipeline for re-running the R-script.

## Validation of the SNP markers set by KASP assays

The 22 SNP markers selected from *in silico* data were converted in KASP locus specific assays and validated by genotyping 44 rice varieties, the 27 accessions initially used for selecting the panel and 17 further varieties cultivated in Spain and Italy.

For each variety, DNA was isolated from 1 dehusked seed that was set on a 2 mL eppendorf tube with 5–6 solid-glass beads, and partially homogenized using a TissueLyser (Qiagen) for 2 min at 30 Hz. After that, 160 μL of SDS extraction buffer (Tris-HCl 100 mM, Na-EDTA 50 MM, NaCl 500 mM, SDS 1.25 w/v %; pH 8.0) and 1 μL of Proteinase K (20 mg/mL Thermo Scientific) were added to each tube, and the mix was incubated at 50 ˚C for 1 h. After that, precipitation was performed using 50 μL of 3M potassium acetate and incubating samples on ice for 10 minutes. After centrifugation at 13000 rpm for 10 min at 4 ˚C, the supernatant was transferred to a new tube and precipitated with 2.5 vol of ethanol absolute. After centrifugation (13000 rpm x 10 min, 4 ˚C), the supernatant was discarded and the DNA pellet was resuspended in 50 μL of ultra-pure water Milli-Q®. DNA was quantified with a Nanodrop Spectrophotometer.

To design KASP primers specific to each SNP, flanking sequences spanning 60 bp on each side of the targeted variant were obtained from Nipponbare reference genome [19]. Genotyping was performed according to LGC instructions for the KASP™ system (LGC Genomics), preparing each reaction with 50 ng of DNA template, 5 μL of 2x KASP reaction mix, 0.14 μL Assay Mix containing 12 μM of each forward primer and 30 μM reverse primer, in a total volume of 10 μL. Assays were performed in duplicate in 96 well plates. Amplification was performed using a Mastercycler EP Gradient S® (Eppendorf) under the following conditions: 94˚C for 15 minutes; 10 cycles of 94˚C for 20s, 61–55˚C for 60s (dropping 0.6˚C per cycle); 26 cycles of 94˚C for 20s, 55˚C for 60s. Some markers needed a recycling step of 3 or 5 cycles of 94˚C for 20s; 57˚C 60 s to increase definition in the genotype clustering. Fluorescence was

detected using a FLUOR Omega SNP microplate reader (BMG LabTech). Genotype calling was performed using the software package KlusterCaller™ (LGC) (www.lgcgroup.com/software).The results were manually curated for allele calling. The same assays were performed in a different lab using new DNA samples and a StepOne Plus real-time thermocycler (Applied Biosystems, Waltham, USA), to assess the reliability of KASP assays.

### Marker polymorphism analysis

Considering that rice is a self-pollination crop, thus, all genes are predominantly homozygous, the polymorphic information content (PIC) of each SNP was calculated according to the following formula,

$$PIC = 1 - \ (p^2 + \ q^2)$$

where p and q are the frequency of two alleles in a given SNP site [20].

## Results

### Selection of a minimum set of markers

To find a minimal set of molecular markers sufficient enough to discriminate among rice varieties we considered a set of 2094 SNPs that was used previously to fingerprint a collection of 217 cultivars, mainly temperate *japonica* [15]. From this genotypic dataset we used the allelic profiles of 210 rice accessions for identifying those SNPs that discriminate among them. A bioinformatics approach, an R-based algorithm, was used to select by iteration a specific combination of markers for discriminating each variety. After removing SNPs that do not distinguish among these varieties and those with missing alleles, and by pairwise comparison of all genotypes, the algorithm identified 5 combinations of 31 SNP markers that could discriminate the varieties included in the collection (S5 Table). From these 5 sets, the R pipeline recognized a total of 139 redundant SNPs that constitute alternative choices to design KASP probes (S4 Table), and therefore to select those markers for which the KASP assay shows best performance and reproducibility. After checking the number of different genotypes produced by all markers, 4 pairs of varieties from the initial dataset were undistinguishable: the Spanish cultivars Fonsa and Pinyana, the pair Guadiamar and M103, as well as Puntal and Clavel and also the cultivars M-5 and M-9 (S3 Table).

To find a specific combination of markers for discriminating Spanish varieties, from this genotypic dataset we used the allelic profiles of 27 rice accessions among the 44 that are currently cultivated in Spain (S1 Table) for identifying those SNPs that discriminate among them. The R-based algorithm, was used. After removing SNPs that do not distinguish among these 27 varieties and those with missing alleles, and by pairwise comparison of all genotypes, the algorithm identified 780 SNP markers that combined generated 94 sets of 6 markers that could discriminate the 27 rice varieties (S7 Table). From these 94 sets, the R pipeline recognized a total of 249 groups of redundant SNPs (S8 Table). As the algorithm runs by iteration, selecting one marker as the starting point for the next step, some markers were highly prevalent in different sets (ie. chr4_34890139 or chr5_876463). After checking the number of different genotypes produced by all markers, two undistinguishable varieties were again detected, the cultivars Fonsa and Pinyana, which were in fact undistinguishable using the initial panel of 2094 markers.

### Validation of the minimum set of markers

From this list of 780 markers, we selected a group of 22 markers which were more frequently tagged by the algorithm, and designed KASP forward and reverse primers for validating this

genotyping protocol in the 27 rice varieties. First, DNA was isolated from kernels to ensure that the method is useful to discriminate varieties in the absence of a grown plant, as is the case of seeds for certification. Second, to investigate whether this method was also valid to identify varieties and to assess purity in rice sachets, we extracted DNA from single polished grains and could obtain enough DNA to proceed with KASP assays (data no shown). In the preliminary assays, some of the primers failed in producing signal, and were substituted by others from the corresponding group of redundant SNPs to obtain 22 KASP assays with high level of reproducibility and good allele call (Table 1, S6 Table). In this sense, the rice genotyping assays were performed in two different laboratories (at IVIA and at the University of Zaragoza). All tested varieties were homozygous in all loci, except Guadiagran that was heterozygous in the locus corresponding to marker 3_11088482. We estimated the Polymorphic Information Content (PIC) for each marker, that measures the ability of a marker to detect polymorphisms in a set of unrelated genotypes [20], and obtained values between 0.30 and 0.50, indicating that each allele was highly present in the 27 studied varieties (Table 1).

The following step was to apply this set of KASP assays to genotype other 17 rice varieties that were not included in the initial collection to reach a total of 44 varieties (S1 Table), 36 of them corresponding to cultivars currently used in Spain which account for 96% of the certified seed produced in the past season (www.mapa.gob.es). The remaining 9 varieties were cultivars interesting for breeding purposes. The 44 varieties were *japonica* types mostly generated by Spanish breeders from genotypes adapted to the local agroclimatic conditions, therefore they are probably genetically related. The detected polymorphisms in 22 loci allowed the discrimination of 43 SNP profiles, since two varieties, Hispalong and Puntal, were not distinguished

**Table 1. Allele frequency of the 22 selected SNP markers and PIC values for 27 Spanish rice varieties.**

| SNPs | AA | GG | CC | TT | PIC |
|---|---|---|---|---|---|
| 1–504463 | 13 | 14 | 0 | 0 | 0.50 |
| 1–9010958 | 11 | 16 | 0 | 0 | 0.48 |
| 1–21657807 | 10 | 17 | 0 | 0 | 0.47 |
| 1–25253276 | 12 | 0 | 15 | 0 | 0.49 |
| 1–26375603 | 5 | 22 | 0 | 0 | 0.30 |
| 1–27325849 | 8 | 19 | 0 | 0 | 0.42 |
| 3–11088482 | 14 | 0 | 0 | 13 | 0.50 |
| 4–503727 | 12 | 15 | 0 | 0 | 0.49 |
| 4–2500045 | 6 | 0 | 21 | 0 | 0.35 |
| 4–34890139 | 12 | 15 | 0 | 0 | 0.49 |
| 5–361719 | 13 | 0 | 14 | 0 | 0.50 |
| 5–876463 | 14 | 13 | 0 | 0 | 0.50 |
| 5–6874605 | 14 | 13 | 0 | 0 | 0.50 |
| 5–7748249 | 13 | 14 | 0 | 0 | 0.50 |
| 5–9000941 | 15 | 12 | 0 | 0 | 0.49 |
| 6–11249268 | 14 | 13 | 0 | 0 | 0.50 |
| 6–12951899 | 13 | 14 | 0 | 0 | 0.50 |
| 6–13355270 | 18 | 0 | 0 | 9 | 0.44 |
| 8–22836510 | 17 | 0 | 10 | 0 | 0.47 |
| 10–875427 | 16 | 11 | 0 | 0 | 0.48 |
| 11–2960303 | 8 | 19 | 0 | 0 | 0.42 |
| 12–23125143 | 11 | 0 | 16 | 0 | 0.48 |

(Table 2). Results obtained in two independent facilities were consistent and minor discrepancies, 1.3%, were found (S9 Table).

When the R-pipeline was applied to this set of 44 rice varieties, the algorithm found 29 combinations of 8 SNP markers (Table 3) that were able to discriminate among the genetic profiles determined. Therefore, any of these sets of 8 SNP-based KASP assays can be used for molecular identification of most of the rice varieties produced in Spain.

## Discussion

In this study, we have developed a simple and easy-to-use bioinformatics tool based on the widely used and freely available software R to facilitate the identification of molecular markers useful for discriminating plant varieties. We have applied this tool to a SNP dataset of temperate *japonica* rice varieties and found 5 sets of 31 markers to discriminate 210 varieties. This number of markers is in tune of other reports as, for example, differentiation of 368 Indian wheat varieties by 54 SNPs [10] or 48 SNP markers to discriminate 518 rice varieties [17]. As our research is focused on the identification of Spanish rice varieties, as there is an increasing demand on methods to protect breeders' rights and food industry traceability requirements, we have used this R-based algorithm as a proof of concept to discriminate these varieties, and also as a transfer of research-derived knowledge to the rice market agents: breeders, producers, manufacturers, regulators, and consumers. Extensive breeding activities during years have narrowed the Spanish rice genetic pool in a way that, frequently, it is difficult to discern between varieties according only to plant morphological features and a detailed analysis of the phenotype is needed. This is never a trivial question for certified seed production, but it is particularly relevant when seeds and grain from some traditional varieties such as Bomba can reach prizes varying 2,5-3- fold those of other cultivars, and in a context of increasing consumers' demand of monovarietal packages of varieties such as JSendra or Albufera (these 3 varieties account for almost 28% of the certified seed in Spain). Therefore, in these cases the need of assuring varietal identity and purity become even more interesting, and a supplementary DNA based strategy, such as SNP genotyping, would be useful for both rice breeders and producers.

To find a minimal set of SNP markers able to discriminate Spanish rice varieties, we used a dataset of 2094 SNPs that was generated by genotyping a temperate *japonica* rice collection [15]. We started from 27 varieties that were genotyped with this panel to identify 22 markers easier to use and cheaper for variety discrimination, and we designed KASP assays that were validated in these varieties and tested in other 17, recently generated varieties. Two pairs of varieties were undistinguishable: Fonsa and Pinyana that displayed identical genetic profile for the initial set of 2094 SNPs, and Hispalong and Puntal, that showed the same alleles for the 22 selected markers. In contrast, Bomba and Bombón, two varieties that are closely related, could be distinguished by more than one SNP. Both varieties are landraces that have been cultivated in Spain since as early as the 19th century, and belong to the same genetic group [15].

KASP markers have been used for variety identification of several plant species, including rice. However, this method may find difficulties in discriminating very close genetically related varieties such as mutants, genome edited plants or those carrying introgressed genes, that could be considered as derivatives of a variety, where the genetic relationship is very close and the differences are few. In our case, as mentioned, it was not possible to distinguish between two pairs of Spanish varieties, probably because they came from the same breeding program and may fall in the mentioned category. In addition, we found four pairs of varieties in the collection that were undistinguishable using 1317 SNPs. In this sense, Owen et al. (2019) found seven pairs among 700 barley varieties that were not distinguished across 6138 SNPs. Two pairs consisted of varieties very different to each other, while one pair was formed by two

**Table 2. Allelic profiles of 44 japonica rice.**

| Variety | 1-504463 | 1-9010958 | 1-21657807 | 1-25253276 | 1-26375603 | 1-27325849 | 3-11088482 | 4-503727 | 4-2500045 | 4-34890139 | 5-361719 | 5-876463 | 5-6874605 | 5-7748249 | 5-9000941 | 6-11249268 | 6-12951899 | 6-13355270 | 8-22836510 | 10-875427 | 11-2960303 | 12-23125143 |
|---|---|---|---|---|---|---|---|---|---|---|---|---|---|---|---|---|---|---|---|---|---|---|
| Albufera | TT | AA | CC | CC | TT | TT | AA | GG | CC | TT | GG | TT | GG | AA | TT | GG | AA | TT | AA | GG | GG | GG |
| Antara | TT | GG | TT | AA | CC | CC | TT | GG | AA | CC | TT | TT | AA | AA | TT | AA | AA | AA | CC | GG | GG | TT |
| Argila | TT | GG | TT | AA | CC | CC | TT | GG | AA | TT | TT | CC | GG | GG | CC | AA | AA | AA | CC | GG | GG | TT |
| Arrodelta | CC | GG | CC | CC | CC | CC | AA | GG | CC | TT | TT | CC | GG | GG | CC | AA | GG | tt | AA | AA | GG | TT |
| Arpa | CC | AA | CC | CC | CC | CC | TT | AA | CC | CC | TT | TT | GG | GG | CC | AA | GG | AA | AA | GG | AA | TT |
| Bahia | CC | GG | CC | CC | CC | CC | TT | AA | CC | CC | GG | TT | AA | AA | TT | GG | AA | TT | AA | GG | GG | TT |
| BalillaxSollana | CC | AA | CC | CC | CC | CC | TT | AA | CC | CC | GG | CC | GG | GG | CC | GG | GG | AA | AA | AA | AA | TT |
| Bomba | CC | AA | CC | AA | TT | TT | AA | GG | CC | tt | GG | CC | GG | GG | TT | GG | GG | AA | AA | AA | GG | GG |
| Bombón | CC | AA | CC | CC | TT | TT | AA | GG | CC | TT | TT | TT | GG | GG | TT | AA | GG | TT | AA | AA | GG | TT |
| Carnaroli | CC | GG | TT | CC | CC | CC | AA | GG | CC | TT | GG | TT | AA | GG | TT | GG | GG | TT | AA | AA | AA | GG |
| Copsemar7 | TT | GG | TT | AA | CC | CC | TT | GG | AA | CC | GG | CC | AA | AA | TT | AA | AA | AA | CC | GG | GG | TT |
| Copsemar8 | CC | AA | CC | CC | CC | CC | TT | GG | CC | TT | GG | CC | GG | GG | CC | GG | GG | AA | AA | AA | GG | GG |
| Copsemar9 | CC | GG | TT | CC | CC | CC | tt | GG | AA | TT | GG | CC | GG | GG | CC | AA | GG | AA | AA | GG | GG | TT |
| Fado | CC | GG | CC | CC | TT | CC | TT | GG | CC | TT | GG | CC | GG | GG | CC | GG | GG | TT | CC | GG | GG | GG |
| Fleixa | TT | GG | CC | CC | TT | TT | AA | AA | CC | TT | TT | TT | AA | AA | CC | AA | AA | AA | CC | AA | GG | GG |
| Fonsa | TT | GG | TT | AA | CC | CC | TT | AA | CC | TT | GG | CC | AA | AA | TT | AA | AA | AA | AA | AA | AA | GG |
| Furia CL | CC | GG | CC | CC | CC | CC | TT | AA | CC | TT | GG | CC | GG | GG | CC | AA | GG | AA | AA | AA | GG | TT |
| Garbell | TT | GG | CC | CC | CC | CC | AA | GG | AA | TT | TT | TT | AA | AA | TT | GG | GG | TT | AA | AA | GG | TT |
| Gavina | CC | GG | CC | CC | TT | TT | AA | AA | CC | CC | TT | TT | AA | AA | TT | AA | GG | TT | AA | AA | GG | GG |
| Gleva | TT | GG | CC | AA | CC | CC | AA | AA | CC | CC | GG | CC | AA | AA | TT | GG | GG | TT | AA | AA | AA | TT |
| Guadiagran | CC | AA | CC | AA | TT | TT | at | GG | ca | TT | GG | CC | GG | GG | CC | GG | GG | AA | AA | GG | AA | TT |
| Guadiamar | CC | AA | CC | AA | CC | CC | TT | GG | AA | TT | GG | CC | GG | GG | CC | GG | GG | AA | CC | gg | AA | TT |
| Guara | TT | GG | CC | aa | CC | CC | AA | aa | AA | TT | TT | TT | AA | GG | TT | GG | GG | TT | ca | AA | GG | TT |
| Hispagran | CC | AA | CC | CC | CC | CC | TT | AA | CC | TT | GG | CC | GG | GG | CC | AA | AA | AA | AA | GG | GG | TT |
| Hispalong | CC | GG | CC | CC | CC | CC | AA | GG | CC | TT | TT | CC | GG | GG | CC | AA | GG | TT | AA | AA | GG | GG |
| Hispamar | CC | GG | CC | AA | CC | CC | AA | gg | AA | TT | GG | CC | AA | GG | CC | GG | GG | AA | AA | AA | GG | GG |
| Jsendra | TT | AA | TT | CC | CC | TT | TT | GG | AA | TT | TT | TT | AA | AA | TT | AA | GG | TT | CC | GG | GG | TT |
| Lido | CC | GG | CC | CC | CC | CC | AA | GG | CC | TT | TT | TT | AA | AA | TT | GG | AA | AA | AA | GG | AA | GG |
| Lluent | CC | GG | CC | CC | CC | CC | AA | AA | CC | TT | GG | CC | GG | GG | CC | AA | GG | AA | AA | GG | GG | GG |
| Luna CL | AA | AA | CC | CC | CC | CC | AA | GG | CC | TT | GG | CC | GG | GG | CC | AA | GG | TT | AA | AA | GG | GG |
| Mare CL | CC | GG | CC | AA | CC | CC | AA | gg | AA | TT | GG | CC | AA | GG | CC | GG | GG | AA | AA | GG | GG | GG |
| Marisma | TT | AA | CC | AA | CC | TT | TT | AA | AA | TT | TT | TT | AA | AA | TT | AA | GG | TT | CC | GG | GG | TT |
| Montsianell | TT | AA | TT | CC | TT | TT | AA | GG | CC | TT | GG | TT | AA | AA | TT | GG | AA | AA | AA | AA | GG | GG |
| Nemesi CL | CC | AA | TT | TT | TT | CC | AA | AA | CC | CC | TT | TT | GG | GG | CC | AA | GG | TT | AA | GG | GG | TT |
| N. Maratelli | CC | AA | CC | CC | TT | TT | AA | GG | CC | TT | GG | TT | AA | AA | CC | AA | GG | AA | AA | AA | GG | GG |
| Olesa | CC | AA | CC | CC | CC | CC | AA | GG | CC | TT | GG | CC | AA | AA | CC | AA | AA | AA | AA | GG | GG | GG |
| Pinyana | TT | GG | TT | AA | CC | CC | TT | AA | AA | CC | GG | TT | AA | TT | TT | AA | AA | TT | AA | AA | AA | GG |
| Puntal | CC | GG | CC | CC | CC | CC | AA | AA | CC | CC | GG | CC | AA | AA | CC | AA | AA | AA | AA | AA | AA | GG |
| Regina | CC | GG | CC | CC | CC | CC | TT | GG | CC | TT | GG | CC | GG | GG | CC | GG | GG | AA | CC | GG | GG | GG |
| Ricastello | TT | GG | TT | AA | TT | CC | AA | AA | CC | TT | TT | AA | AA | AA | TT | AA | AA | AA | AA | AA | AA | GG |
| Sirio CL | CC | GG | CC | CC | CC | CC | AA | GG | CC | TT | GG | CC | GG | GG | CC | GG | GG | AA | AA | GG | GG | GG |
| Soto | TT | GG | TT | CC | TT | TT | TT | AA | CC | TT | GG | CC | GG | GG | CC | AA | AA | AA | CC | GG | GG | GG |
| Tebre | TT | AA | TT | CC | TT | TT | AA | AA | CC | CC | TT | TT | AA | AA | TT | GG | GG | AA | AA | AA | GG | GG |
| Thaiperla | CC | AA | CC | AA | CC | CC | TT | AA | CC | TT | GG | CC | GG | GG | CC | aa | AA | AA | CC | GG | GG | TT |

Allelic profiles determined for 22 SNP markers by KASP assays performed in two independent facilities. Genotype discrepancies between assays are noted in low-case letters.

varieties with very similar phenotypes [7]. Tang et al. selected a panel of 48 SNP markers to discriminate 518 rice varieties and they found pairs of *japonica* varieties that couldn't be discriminate from each other using the 48 KASP markers [17]. These varieties came from the same breeding institutes, may share the same parents and, consequently, they show a high genetic background in common.

The developed algorithm found that this resolution can be achieved by using only sets of 8 SNP markers, that were provided by the R-based pipeline. Therefore, a simple, rapid, and cost-effective PCR-based protocol was established to discriminate most of the rice varieties currently cultivated in Spain, despite of their close relationship. The availability of this temperate *japonica* rice genotyping dataset enables the addition of new varieties as they are being released, that could be genotyped using this information as reference. The algorithm described in this study would also be suitable for identifying minimal sets of discriminating markers useful for other groups of plant varieties. The algorithm also recognizes redundant markers which are particularly valuable to design efficient and reliable KASP markers with low fail and error rates. In our study, DNA isolation and KASP assays were performed in duplicate at two

**Table 3. List of sets of 8 SNP markers for discriminating 44 *japonica* rice varieties.** SNP are named by their chromosome number followed by their position.

| Set | marker 1 | marker 2 | marker 3 | marker 4 | marker 5 | marker 6 | marker 7 | marker 8 |
|---|---|---|---|---|---|---|---|---|
| 1 | 1_9010958 | 3_11088482 | 4_34890139 | 6_13355270 | 5_361719 | 6_12951899 | 8_22836510 | 6_11249268 |
| 2 | 1_9010958 | 3_11088482 | 4_34890139 | 6_13355270 | 5_361719 | 8_22836510 | 1_21657807 | 6_11249268 |
| 3 | 1_9010958 | 3_11088482 | 4_34890139 | 6_13355270 | 5_876463 | 8_22836510 | 1_21657807 | 6_11249268 |
| 4 | 1_9010958 | 3_11088482 | 5_876463 | 6_13355270 | 8_22836510 | 4_34890139 | 1_21657807 | 6_11249268 |
| 5 | 1_9010958 | 3_11088482 | 4_34890139 | 6_13355270 | 5_361719 | 8_22836510 | 6_12951899 | 6_11249268 |
| 6 | 1_9010958 | 3_11088482 | 4_34890139 | 4_503727 | 5_876463 | 1_25253276 | 6_13355270 | 1_504463 |
| 7 | 1_9010958 | 3_11088482 | 5_876463 | 4_503727 | 1_25253276 | 6_13355270 | 4_34890139 | 1_504463 |
| 8 | 1_9010958 | 3_11088482 | 5_876463 | 6_13355270 | 1_25253276 | 4_503727 | 4_34890139 | 1_504463 |
| 9 | 1_9010958 | 3_11088482 | 4_34890139 | 6_13355270 | 5_876463 | 1_25253276 | 4_503727 | 1_504463 |
| 10 | 1_9010958 | 3_11088482 | 5_876463 | 6_13355270 | 1_25253276 | 4_34890139 | 4_503727 | 1_504463 |
| 11 | 1_9010958 | 11_23125143 | 4_34890139 | 6_13355270 | 5_876463 | 4_2500045 | 1_25253276 | 1_26375603 |
| 12 | 1_9010958 | 11_23125143 | 5_876463 | 6_13355270 | 4_2500045 | 4_34890139 | 1_25253276 | 1_26375603 |
| 13 | 1_9010958 | 3_11088482 | 5_876463 | 4_503727 | 1_25253276 | 6_13355270 | 5_7748249 | 1_26375603 |
| 14 | 1_9010958 | 3_11088482 | 5_876463 | 6_13355270 | 1_25253276 | 4_503727 | 5_7748249 | 1_26375603 |
| 15 | 1_9010958 | 3_11088482 | 4_34890139 | 4_503727 | 5_361719 | 8_22836510 | 1_25253276 | 5_9000941 |
| 16 | 1_9010958 | 3_11088482 | 4_34890139 | 4_503727 | 5_361719 | 8_22836510 | 1_25253276 | 6_11249268 |
| 17 | 1_9010958 | 3_11088482 | 4_34890139 | 6_13355270 | 5_361719 | 1_25253276 | 10_875427 | 6_11249268 |
| 18 | 1_9010958 | 3_11088482 | 4_34890139 | 4_503727 | 5_361719 | 8_22836510 | 1_26375603 | 6_11249268 |
| 19 | 1_9010958 | 3_11088482 | 4_34890139 | 6_13355270 | 5_361719 | 1_25253276 | 6_11249268 | 10_875427 |
| 20 | 1_9010958 | 3_11088482 | 4_34890139 | 4_503727 | 5_361719 | 1_25253276 | 1_504463 | 1_21657807 |
| 21 | 1_9010958 | 3_11088482 | 4_34890139 | 4_503727 | 5_361719 | 1_25253276 | 1_504463 | 6_11249268 |
| 22 | 1_9010958 | 3_11088482 | 4_34890139 | 4_503727 | 5_361719 | 1_25253276 | 1_26375603 | 6_11249268 |
| 23 | 1_9010958 | 3_11088482 | 4_34890139 | 4_503727 | 5_361719 | 1_25253276 | 5_9000941 | 1_21657807 |
| 24 | 1_9010958 | 3_11088482 | 4_34890139 | 4_503727 | 5_361719 | 1_25253276 | 5_9000941 | 8_22836510 |
| 25 | 1_9010958 | 3_11088482 | 4_34890139 | 4_503727 | 5_361719 | 1_25253276 | 6_11249268 | 1_504463 |
| 26 | 1_9010958 | 3_11088482 | 4_34890139 | 4_503727 | 5_361719 | 1_25253276 | 6_11249268 | 1_21657807 |
| 27 | 1_9010958 | 3_11088482 | 4_34890139 | 4_503727 | 5_361719 | 1_25253276 | 6_11249268 | 1_26375603 |
| 28 | 1_9010958 | 3_11088482 | 4_34890139 | 4_503727 | 5_361719 | 1_25253276 | 6_11249268 | 8_22836510 |
| 29 | 1_9010958 | 3_11088482 | 4_34890139 | 4_503727 | 5_361719 | 1_25253276 | 6_11249268 | 10_875427 |

facilities and using different equipment for amplification and allele calling and a high consistency in the results was found.

Using DUS tests to identify varieties requires effort and it is time consuming as it involves the phenotyping of a high number of plants from seedling to maturing stage. Variety identification using molecular markers has been recently considered as a complement in the variety distinction protocols for species marketed in the European Union (EU Implementing Directive 2021/971). But its reliability and reproducibility make it worthy of being considered as one of the main tests for the identification of varieties.

## Conclusions

We have developed a simple and easy-to-use bioinformatics tool using the widely used and freely available software R to facilitate the identification of varieties. Using this algorithm, we found that minimal sets of 8 SNP are enough to discriminate 44 varieties that are currently cultivated in Spain, generating a reference genotype of each variety. This R-based algorithm and the KASP-markers panel for *japonica* rice varieties identification constitute useful tools for breeders, seed certifiers and food inspection agencies.

## Supporting information

**S1 Table. List of *japonica* rice cultivars used in this study.**
(XLSX)

**S2 Table. Script used in this study.**
(TXT)

**S3 Table. List of pairs of non-differentiable accession in the 210 varieties collection using 1317 SNPs.**
(XLSX)

**S4 Table. List of groups of redundant markers for discriminating 210 rice cultivars as derived from genotypes previously obtained.**
(XLSX)

**S5 Table. Sets of SNP markers that distinguished the varieties in the collection.**
(TXT)

**S6 Table. Sequences of the 22 primers used in this study.** The SNPs in each primer are indicated in brackets.
(XLSX)

**S7 Table. List of minimal sets of 6 SNP markers for discriminating 27 rice varieties as derived from a germplasm collection previously genotyped.**
(XLSX)

**S8 Table. List of groups of redundant markers for discriminating 27 rice cultivars as derived from genotypes previously obtained.**
(DOCX)

**S9 Table. Genotyping results from KASP assays performed at two facilities (score 1 from IVIA and score 2 from UZ), and from the initial SNP dataset.**
(XLSX)

## Acknowledgments

We acknowledge Juan Luis Reig-Valiente, who passed away, for his contribution to this article.

## Author Contributions

**Conceptualization:** Ester Sales, Concha Domingo.

**Data curation:** Ester Sales, Julia García-Romeral, Concha Domingo.

**Formal analysis:** Ester Sales, Concha Domingo.

**Funding acquisition:** Julia García-Romeral, Concha Domingo.

**Investigation:** Ester Sales, Concha Domingo.

**Methodology:** Ester Sales, Julia García-Romeral, Concha Domingo.

**Software:** Julia García-Romeral.

**Supervision:** Concha Domingo.

**Validation:** Ester Sales, Julia García-Romeral.

**Visualization:** Julia García-Romeral.

**Writing – original draft:** Concha Domingo.

**Writing – review & editing:** Ester Sales, Concha Domingo.

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
