## [Decision Letter · Decision Letter 0]

11 Apr 2023

PONE-D-23-04509Bioinformatics approach for developing a minimum set of SNP markers for identification of temperate japonica rice varieties cultivated in SpainPLOS ONE

Dear Dr. Domingo,

Thank you for submitting your manuscript to PLOS ONE. After careful consideration, we feel that it has merit but does not fully meet PLOS ONE’s publication criteria as it currently stands. Therefore, we invite you to submit a revised version of the manuscript that addresses the points raised during the review process.

Authors should address the reviewer's comments and critically discuss the results, especially  why  some of the genotypes could not be separated in clusters as per their geographical distribution?.  The results would have been of more practical relevance had they included rice varieties from other countries as well.

We look forward to receiving your revised manuscript.

Kind regards,

Amit Kumar Singh

Academic Editor

PLOS ONE

Journal Requirements:

"JGR was funded by the Spanish Ministerio de Ciencia, Innovación y Universidades (https://www.ciencia.gob.es/) (fellowship number PRE2019-089034)"

Additional Editor Comments:

Reviewers have found this study interesting, however they some concerns which need to be addressed by the authors.

Reviewers' comments:

Reviewer's Responses to Questions

**Comments to the Author**

1. Is the manuscript technically sound, and do the data support the conclusions?

Reviewer #1: Yes

Reviewer #2: Yes

2. Has the statistical analysis been performed appropriately and rigorously? 

Reviewer #1: Yes

Reviewer #2: Yes

3. Have the authors made all data underlying the findings in their manuscript fully available?

Reviewer #1: Yes

Reviewer #2: Yes

4. Is the manuscript presented in an intelligible fashion and written in standard English?

Reviewer #1: Yes

Reviewer #2: Yes

5. Review Comments to the Author

Reviewer #1: The present study has been done to develop an assay with a minimum number of SNP markers that can distinguish japonica rice varieties grown in Spain. For this purpose, 44 japonica rice varieties were studied with 22 KASP markers. For the selection of these distinguishing KASP markers a freely available software R was used to generate small sets of SNPs that can discriminate these varieties, by selecting markers from a larger genotyping dataset. Based on the data generated with 22 KASP markers on 44 rice varieties further R package-based algorithm identified 8 KASP markers which were sufficient to distinguish japonica rice varieties. I need a few clarifications based on the results presented by the authors

1. What was the reason that Hispalong and Puntal varieties could not get distinguished based on 22 KASP markers assay developed? This needs to be elaborated.

2. What additional effort was done to separate Hispalong and Puntal before proposing the minimal marker set (8 SNPs), this issue needs to be addressed.

3. Line no. 25 needs to be corrected because the set of 8 SNP markers could distinguish 43 varieties instead of 44.

4. This minimal set (8 SNPs) of SNP identified has very limited applicability because they can distinguish only 42 japonica rice varieties grown in Spain. This needs to be tested on another set of japonica varieties (from other countries) to prove its wider applicability.

Reviewer #2: The manuscript entitled "Bioinformatics approach for developing a minimum set of SNP markers for identification of temperate japonica rice varieties cultivated in Spain" by Sales et al. demonstrates the use of molecular markers (SNP/ small and low-cost SNP panels) for rice variety identification and protection. They developed a user friendly bioinformatics algorithm in R software for varieties differentiation by selecting markers from a larger genotyping dataset. They applied this algorithm to discriminated the 44 widely cultivated Spain rice varieties with minimal sets of 8 SNPs. They also developed and tested 22 KASP assays for the markers included in these panels, and obtained reliable genotype patterns for rice varieties identification. The work is interesting and of greater benefits to rice breeders, seed certifiers and food inspection agencies. I recommend its publication but request the following queries/ comments to be incorporated before acceptance:

Introduction:

Line 44: Comma after "Italy"

Lines 46-48: Reframe for meaningful sentence.

Materials and methods

Line 55-58: "However, this phenotype-based ... programs are developed." needs revision.

Line 138: Italicise "in silico" and wherever it occurs.

Results

Table 3: The markers mentioned, what it stands should be captioned in the legend (like chromosomenumber_postion) for clarity to readers.

The proper results for "Marker polymorphism analysis" is missing in "Results"

Discussion

Discussion in general may be further expanded/ elaborated.

Following paper may be cited in view of the application of the presented work: https://www.nature.com/articles/s41598-019-41204-2

General comment:

Authors claim that the study attends the need of developing an efficient molecular tool for the identification/ differentiation of varieties. https://www.ncbi.nlm.nih.gov/pmc/articles/PMC7779600/. The work is impressive but the R package should be submitted/ published in CRAN for the general users/researchers once it gets accepted.

6. PLOS authors have the option to publish the peer review history of their article (what does this mean?). If published, this will include your full peer review and any attached files.

Reviewer #1: **Yes: **Rakesh Singh

Reviewer #2: **Yes: **Sarika Jaiswal

Division of Agricultural Bioinformatics

ICAR-Indian Agricultural Statistics Research Institute

Library Avenue, PUSA, New Delhi -110012 , INDIA

---

## [Author Response · Author response to Decision Letter 0]

10 May 2023

We appreciate reviewer’s comments since they have made it possible to improve the manuscript and make it more understandable

Answers to reviewers' comments:

Reviewer #1: The present study has been done to develop an assay with a minimum number of SNP markers that can distinguish japonica rice varieties grown in Spain. For this purpose, 44 japonica rice varieties were studied with 22 KASP markers. For the selection of these distinguishing KASP markers a freely available software R was used to generate small sets of SNPs that can discriminate these varieties, by selecting markers from a larger genotyping dataset. Based on the data generated with 22 KASP markers on 44 rice varieties further R package-based algorithm identified 8 KASP markers which were sufficient to distinguish japonica rice varieties. I need a few clarifications based on the results presented by the authors

1. What was the reason that Hispalong and Puntal varieties could not get distinguished based on 22 KASP markers assay developed? This needs to be elaborated.

The use of molecular markers sets for identification is limited. As cited in the text, there are several reported cases in which varieties are not distinguishable using large sets of markers (Owen, Genet Resour Crop Evol 2019; Tang, Rice. 2022). Hispalong is a derived variety from Puntal in which genes have been introgressed. The information of the genes that have been introgressed is confidential and it is kept by the breeders. In addition, in the new version, we have included the analysis of 210 varieties from different countries using 2094 markers and found that 4 pairs of varieties were indistinguishable. 

2. What additional effort was done to separate Hispalong and Puntal before proposing the minimal marker set (8 SNPs), this issue needs to be addressed.

The genotyped both Hispalong and Puntal using 44 molecular markers. We also cultivated Hispalong and Puntal plants and we found that they were morphologically different.

3. Line no. 25 needs to be corrected because the set of 8 SNP markers could distinguish 43 varieties instead of 44.

We appreciate the reviewer’s comment. This has been changed in the manuscript.

4. This minimal set (8 SNPs) of SNP identified has very limited applicability because they can distinguish only 42 japonica rice varieties grown in Spain. This needs to be tested on another set of japonica varieties (from other countries) to prove its wider applicability.

The aim of this study was to develop an algorithm that could be used to develop different sets of markers depending on the varieties to analysed. This algorithm can be used with any SNP dataset. In our case we started from a set of 2094 markers obtained from temperate japonica varieties and it should be useful for this genetic population. In the case that a different of varieties should be analysed, the algorithm will rise a set of different markers. As a proof of concept, we applied the algorithm to discriminate Spanish varieties and found that 8 SNPs were needed to distinguish 27 varieties and 8 to discriminate 42.

Following reviewer comment, we have added a list of markers needed to discriminate the 210 varieties included in the japonica temperate collection, using the algorithm 

Reviewer #2: The manuscript entitled "Bioinformatics approach for developing a minimum set of SNP markers for identification of temperate japonica rice varieties cultivated in Spain" by Sales et al. demonstrates the use of molecular markers (SNP/ small and low-cost SNP panels) for rice variety identification and protection. They developed a user friendly bioinformatics algorithm in R software for varieties differentiation by selecting markers from a larger genotyping dataset. They applied this algorithm to discriminated the 44 widely cultivated Spain rice varieties with minimal sets of 8 SNPs. They also developed and tested 22 KASP assays for the markers included in these panels, and obtained reliable genotype patterns for rice varieties identification. The work is interesting and of greater benefits to rice breeders, seed certifiers and food inspection agencies. I recommend its publication but request the following queries/ comments to be incorporated before acceptance:

Introduction:

- Line 44: Comma after "Italy"

This has been modified in the revised manuscript.

- Lines 46-48: Reframe for meaningful sentence.

This has been modified in the revised manuscript.

Materials and methods

- Line 55-58: "However, this phenotype-based ... programs are developed." needs revision.

This sentence has been revised in the new version of the manuscript.

- Line 138: Italicise "in silico" and wherever it occurs.

This has been modified in the revised manuscript.

Results

- Table 3: The markers mentioned, what it stands should be captioned in the legend (like chromosomenumber_postion) for clarity to readers.

This is now indicated in the revised manuscript.

- The proper results for "Marker polymorphism analysis" is missing in "Results"

We don’t understand this comment, "Marker polymorphism analysis" is indicated in line 228-230 and Table 1.

Discussion

- Discussion in general may be further expanded/ elaborated.

Following paper may be cited in view of the application of the presented work: https://www.nature.com/articles/s41598-019-41204-2

Discussion has been revised and the mentioned manuscript has been cited (page 4, line 80; page 16, line 262)

General comment:

- Authors claim that the study attends the need of developing an efficient molecular tool for the identification/ differentiation of varieties. https://www.ncbi.nlm.nih.gov/pmc/articles/PMC7779600/. 

This mentioned manuscript has been cited (page 5, line 83)

- The work is impressive but the R package should be submitted/ published in CRAN for the general users/researchers once it gets accepted.

The R-package has been submitted to GitHub with the name “minimal-markers” and it is publicly available. The script is also included in the manuscript in Supplementary file 2

---

## [Decision Letter · Decision Letter 1]

24 May 2023

Bioinformatics approach for developing a minimum set of SNP markers for identification of temperate japonica rice varieties cultivated in Spain

PONE-D-23-04509R1

Dear Dr. Domingo

We’re pleased to inform you that your manuscript has been judged scientifically suitable for publication and will be formally accepted for publication once it meets all outstanding technical requirements.

Kind regards,

Amit Kumar Singh

Academic Editor

PLOS ONE

Additional Editor Comments (optional):

Reviewers' comments:

Reviewer's Responses to Questions

**Comments to the Author**

1. If the authors have adequately addressed your comments raised in a previous round of review and you feel that this manuscript is now acceptable for publication, you may indicate that here to bypass the “Comments to the Author” section, enter your conflict of interest statement in the “Confidential to Editor” section, and submit your "Accept" recommendation.

Reviewer #1: All comments have been addressed

2. Is the manuscript technically sound, and do the data support the conclusions?

Reviewer #1: Yes

3. Has the statistical analysis been performed appropriately and rigorously? 

Reviewer #1: Yes

4. Have the authors made all data underlying the findings in their manuscript fully available?

Reviewer #1: Yes

5. Is the manuscript presented in an intelligible fashion and written in standard English?

Reviewer #1: Yes

6. Review Comments to the Author

Reviewer #1: The authors have addressed all the comments therefore the manuscript may be accepted for the publication.

7. PLOS authors have the option to publish the peer review history of their article (what does this mean?). If published, this will include your full peer review and any attached files.

Reviewer #1: **Yes: **Dr Rakesh Singh

---

## [Editor Report · Acceptance letter]

14 Jun 2023

PONE-D-23-04509R1 

Bioinformatics approach for developing a minimum set of SNP markers for identification of temperate *japonica* rice varieties cultivated in Spain 

Dear Dr. Domingo:

I'm pleased to inform you that your manuscript has been deemed suitable for publication in PLOS ONE. Congratulations! Your manuscript is now with our production department. 

Kind regards, 

on behalf of

Dr Amit Kumar Singh 

Academic Editor

PLOS ONE